# Chiral phonons in polar LiNbO₃

Hiroki Ueda [1,7] ✉, Abhishek Nag [1,5,7] ✉, Carl P. Romao [2,3], Mirian García-Fernández [4], Ke-Jin Zhou [4,6] & Urs Staub [1] ✉

Quasiparticles describe collective excitations in many-body systems, and their symmetry classification is of fundamental importance for physical processes such as excited states, transport phenomena, and phase transitions. Recent studies have introduced chirality as an additional degree of freedom in condensed matter physics, leading to a range of novel phenomena. Among these, chiral phonons are of special interest because they carry angular momentum and therefore intrinsically break time reversal symmetry, which non-trivially bridges the spin system with the lattice. Here, we directly prove the presence of chiral phonons in a prototypical polar LiNbO₃ crystal. Our demonstration of chiral phonons in a ferroelectric enables in-situ electrical control of momentum-dependent "magnetic" polarization with the reversible phonon handedness. This ferroic control of phonon chirality has substantial potential in the emerging field of chiral phononics, particularly along the associated control of its phonon angular momentum.

A phonon is an archetypical quasiparticle describing collective atomic motions as a single boson. Even though a phonon usually describes lattice excitations, it has been demonstrated that it can have a character mixed with magnetism, opening the intriguing possibility of phononic control of magnetic moments. If such a phonon is infrared active, it can be classified as a dynamical multiferroic mode[1,2]. Examples include electromagnons[3,4] and phonons with angular momentum[5–8]. The latter is often referred to as a chiral phonon. The magnetic aspect of chiral phonons can be qualitatively understood by the Barnett effect[9] at ultrafast timescales[10]. In equilibrium, the Barnett effect describes the induced magnetization in a spinning magnetically disordered medium. On ultrafast timescales, the magnetization is induced by the revolution of atoms, which could originate either from an intrinsic eigenmode[11,12] or a coherent excitation of degenerate linear modes with a relative phase shift of π/2 driven by a circularly polarized laser pulse at the phonon resonance[2,10,13]. The emergent effective magnetic field from driving chiral phonons has significant potential to control magnetism at ultrafast timescales, e.g., as recently applied to magnetization switching[10] and coherent magnon excitation[14]. The "magnetism" of phonons has also theoretically been investigated[15–20] to form the fundamental basis of the new research field, chiral phononics.

Chiral phonons have attracted further interest from the opposite perspective, i.e., absorbing an angular momentum quantum from the magnetic system, which is not possible for conventional phonons. Recent experimental works have demonstrated that the angular momentum transfer occurs between spins and chiral (rotational) phonons at ultrafast timescales, known as the ultrafast Einstein–de Haas effect[21,22], which is exactly the inverse effect of the ultrafast Barnett effect[10] and is essential for the ultrafast demagnetization process. In addition, the creation of chiral phonons via magnon-phonon conversion highlights the critical role of phonon angular momentum in transport[23,24]. Thermal gradients create phonon angular momentum flow in a chiral crystal due to chiral acoustic phonons[25] and generate a spin current[26]. The substantial spin polarization of electrons propagating through a chiral crystal, known as chirality-induced spin selectivity and reaching an effective magnetic field in the order of 100 T[27], strongly indicates non-trivial coupling between electrons and chiral phonons. Besides, chiral phonons might mediate magnetic exchange interaction in heterostructures[28]. As such, the intrinsic "magnetism" of chiral phonons is spontaneously responsible for spin-lattice coupling and leads to various non-trivial phenomena and functionalities in materials.

[1]Center for Photon Science, Paul Scherrer Institute, Villigen, Switzerland. [2]Department of Materials, ETH Zurich, Zurich, Switzerland. [3]Department of Materials, Faculty of Nuclear Sciences and Physical Engineering, Czech Technical University in Prague, Prague, Czech Republic. [4]Diamond Light Source, Didcot, UK. [5]Present address: Indian Institute of Technology, Roorkee, India. [6]Present address: University of Science and Technology of China, Hefei, Anhui, China. [7]These authors contributed equally: Hiroki Ueda, Abhishek Nag. ✉e-mail: hiroki.ueda@psi.ch; abhishek.nag@ph.iitr.ac.in; urs.staub@psi.ch

Three different types of phonons have been referred to as chiral phonons in the community[29]: (1) a rotational mode at the Γ point, such as the one reported in $SrTiO_3$[2] and $CeF_3$[13], (2) a rotational mode propagating in the rotation plane, such as the mode at the high symmetry points $K$ and $K'$ in transition metal dichalcogenides, e.g., $WSe_2$[8], and (3) a rotational mode that propagates perpendicular to the rotation plane, such as the one observed in chiral crystals, e.g., α-quartz[12], α-HgS[11], and tellurium[30]. All of them possess phonon angular momentum, but the reciprocal lattice volume where these types of phonons exist is significantly larger for the last type compared to the other two because the former two types are confined to a two-dimensional plane, whereas the last type of phonons reside in three-dimensional space. This distinction can lead to different contributions to macroscopic properties. Furthermore, from a symmetry perspective[31], only the last type fulfills the symmetry requirements of being a dynamical chiral object due to these different dimensionalities. Hence, the most relevant and truly chiral phonons have only been experimentally reported in chiral crystals, limiting the exploration of chiral phononics.

Here, we demonstrate the presence of chiral phonons in the prototypical polar crystal $LiNbO_3$ within a single ferroelectric domain by using resonant inelastic X-ray scattering (RIXS) with circular polarization. $LiNbO_3$ was selected for this study for several reasons: (1) it is a well-established prototypical polar crystal, (2) high-quality single-domain crystals are commercially available, (3) ferroelectric domains can be controlled at the nanometer scale[32], and (4) its phonon properties are directly relevant to technological applications, such as substrates for surface acoustic wave devices, thin-film growth, and piezoelectric devices. Angular momentum transfer between a circularly polarized X-ray photon and a chiral phonon needs to fulfill the selection rules of phonon excitation in the RIXS process, which results in a circular polarization contrast on a chiral phonon excitation peak. Density functional theory (DFT) calculations provide reliable phonon properties for $LiNbO_3$ because it contains only a moderate number of non-magnetic light elements in the unit cell without significant electronic correlations. Therefore, combining RIXS experiments and DFT calculations yields an unambiguous and robust demonstration of chiral phonons in a polar (non-chiral) crystal and establishes a large and important class of materials for chiral phononics.

## Results and discussion

$LiNbO_3$ possesses a polar corundum structure with the space group $R3c$ below the Curie temperature 1483 K[33], as shown in Fig. 1b, c (in hexagonal setting). The off-centering of $Li^+$ and $Nb^{5+}$ from the $O^{2-}$ triangle and octahedra, respectively, creates spontaneous polarization along the $c$ axis. Note that $R3c$ is a polar space group without space-inversion symmetry but is not a chiral space group, as it possesses $c$ glide symmetries. Its Brillouin zone viewed along $c^*$ is shown in Fig. 1d, together with the symmetry elements of $R3c$ and some momentum points where we collected RIXS spectra, which include $\mathbf{q}_1 = (0.1, -0.2, 1)$, $\mathbf{q}_2 = (0, -0.175, 1)$, and $\mathbf{q}_3 = (-0.1, -0.1, 1)$. While $\mathbf{q}_2$ is in a glide plane, $\mathbf{q}_1$ and $\mathbf{q}_3$ are interconnected by the glide operation. Measurements on these momentum points allow us to verify circular dichroism (CD) in RIXS of chiral phonon origin based on symmetry arguments, as discussed later.

Chiral phonons with opposite handedness can be expressed by phonon angular momentum $\mathbf{J}$ and phonon momentum $\mathbf{q}$[34], which is either $\mathbf{J} // \mathbf{q}$ (right-handed) or $-\mathbf{J} // \mathbf{q}$ (left-handed). Thus, their inner product is a good quantity to describe phonon chirality[35]. Note that $\mathbf{J}$ is a time-odd axial vector while $\mathbf{q}$ is a time-odd polar vector. They are degenerate at the Γ point as long as the macroscopic time-reversal symmetry is preserved (non-magnetic). In this case, a right-handed mode $(-\mathbf{J}, -\mathbf{q})$ at arbitrary $\mathbf{q}$ is equivalent in energy to a right-handed mode $(\mathbf{J}, \mathbf{q})$. The space-inversion operation connects a right-handed mode $(-\mathbf{J}, -\mathbf{q})$ to a left-handed mode $(-\mathbf{J}, \mathbf{q})$, which makes all chiral phonon pairs $(\mathbf{J}, \mathbf{q})$ and $(-\mathbf{J}, \mathbf{q})$ degenerate at all the momentum points in non-magnetic centrosymmetric crystals. Therefore, space-inversion symmetry breaking is the crucial ingredient for the presence of chiral phonons (see Fig. 1a), i.e., momentum-dependent magnetic polarizations, suggesting the existence of chiral phonons in general non-centrosymmetric crystals and their relevance to a wide class of materials.

Resonant X-ray scattering is described by a second-rank tensor and is sensitive to electric monopole (charge), magnetic dipole (spin), and electric quadrupole (orbital asphericity) for the dominant X-ray scattering process, i.e., electric dipole-electric dipole (E1-E1) transitions[36]. Fig. 2a shows an X-ray absorption spectrum (XAS) around the O K edge. At the O K edge, a $1s$ core electron is excited into a $2p$ shell. Spin contributions are absent in diamagnetic $LiNbO_3$. In this case, CD in RIXS originates from the excitation of O $2p$ electric quadrupoles, as CD requires finite intensities in the polarization rotation channel in the X-ray scattering process, which is absent for isotropic charge scattering. As found in the RIXS energy map shown in Fig. S1b, e in Supplementary Information, phonon resonances are substantial for incident X-ray energies of ~530.85 eV and ~535.25 eV. The latter photon energy (represented by an arrow in Fig. 2a) is chosen for the RIXS measurements, as we do not find a clear CD-RIXS signal at the lower photon energy (compare Fig. S1c and S1f). The highest phonon energy in $LiNbO_3$ is ~110 meV[37], and all features above that energy are due to higher harmonics of phonon excitations.

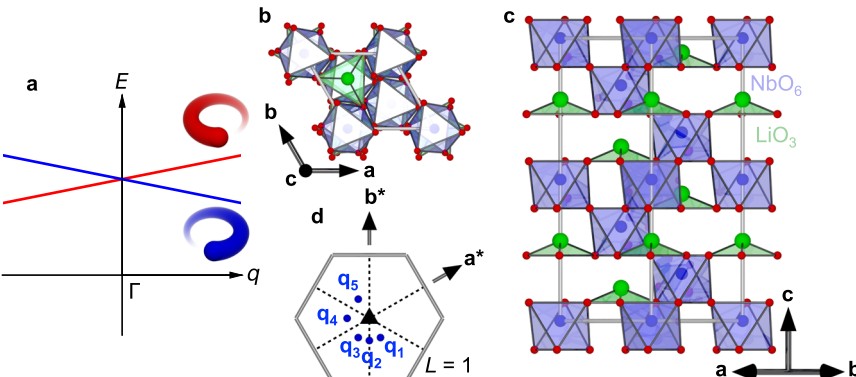

**Fig. 1 | Symmetry requirement of chiral phonons, and crystal structure and Brillouin zone of $LiNbO_3$. a** Schematic drawing of chiral phonon dispersion with broken space-inversion symmetry and preserved time-reversal symmetry. Red and blue lines represent phonon dispersion with opposite phonon angular momenta.

Crystal structure of $LiNbO_3$ in the hexagonal setting viewed along **b** [001] and **c** [110]. **d** Brillouin zone of $LiNbO_3$ viewed along $c^*$ with momentum points where RIXS spectra have been collected. Here, $\mathbf{q}_1 = (0.1, -0.2, 1)$, $\mathbf{q}_2 = (0, -0.175, 1)$, $\mathbf{q}_3 = (-0.1, -0.1, 1)$, $\mathbf{q}_4 = (-0.2, 0.1, 1)$, and $\mathbf{q}_5 = (-0.1, 0.2, 1)$.

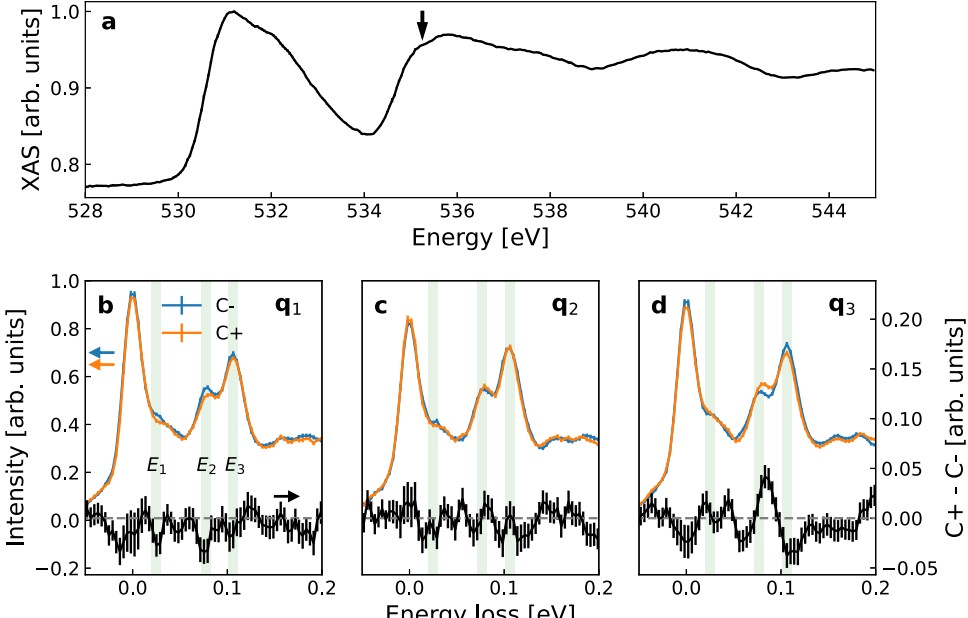

**Fig. 2 | XAS and RIXS. a** XAS around the O $K$ edge. The arrow represents the photon energy for the RIXS measurements. RIXS with circular X-ray polarization at **b $q_1$**, **c $q_2$**, and **d $q_3$**. The green bars highlight the representative energy loss points with finite CD, $E_1$, $E_2$, and $E_3$ (see text). The error bars in an RIXS spectrum are the standard deviation of individual scans.

According to Neumann's principle, chiral phonons in LiNbO$_3$ must respect the symmetry of $R3c$. Since a **J** component along a glide plane flips the sign by the glide operation, chiral phonons are not allowed at **$q_2$** in the glide plane (i.e., **J**$_{//\mathbf{q}} = \mathbf{0}$) but only those with a **J** component normal to the $c$ glide plane (**J**$_{\perp\mathbf{q}} \neq \mathbf{0}$, two-dimensional chiral or cycloidal phonons). In contrast, since **$q_1$** and **$q_3$** are not in a $c$ glide plane but are connected by the glide symmetry, chiral phonons can exist (**J**$_{//\mathbf{q}} \neq \mathbf{0}$) and, if they exist, must reverse their handedness between the two momentum points. In addition, the threefold rotational symmetry along $c$ must be fulfilled. Figure 1d illustrates these relevant symmetry elements in reciprocal space, and Fig. S2a–c represent arrow plots of **J** corresponding to specific phonon modes obtained from DFT calculations (described in "Methods"). Raman spectroscopy with circular polarization probed chiral phonons propagating only along the principal axis, which is the $c$ axis in trigonal crystals, because of the strong birefringence effect[11,30]. However, it does not measure chiral phonons in LiNbO$_3$ because the $c$ axis is in glide planes. RIXS-CD measurements can overcome this limitation by their large momentum accessibility to extend a range of materials relevant for chiral phononics.

Figure 2b–d represents RIXS spectra taken at **$q_1$**, **$q_2$**, and **$q_3$** with circular X-ray polarization (see "Methods" for details). There are three representative energy loss points in the spectra where we can find CD signals: -25 meV ($E_1$), -77 meV ($E_2$), and -106 meV ($E_3$). CD signals are less substantial, e.g., -7% for $E_2$ at **$q_1$**, compared to the one observed in α-quartz, -17%[12]. In comparison to chiral α-quartz, LiNbO$_3$ has additional symmetry elements ($c$ glide planes) that constrain the appearance of nondegenerate chiral phonons, resulting in a generally reduced chiral phonon band splitting. Recent symmetry-based theoretical work categorized chiral phonons in crystals with $C_{3v}$ symmetry as g-wave based on the nodal structure[38].

The RIXS CD is absent at **$q_2$** and is roughly reversed between **$q_1$** and **$q_3$**, as expected from the symmetry analysis, except for $E_3$. As described in detail in the Supplementary Information, the CD signal at $E_3$ seems significantly affected by the X-ray birefringence effect[39,40] because the modes at $E_3$ are almost pure linear translational and have a large mode effective charge (see Fig. S6), resulting in a substantially linear dichroic RIXS amplitude (see Fig. S7). For the case here, the threefold symmetry of the lattice and the twofold symmetry of the

oscillatory electromagnetic waves make the birefringence sixfold symmetric. This results in identical birefringence originated CD between **$q_1$** and **$q_3$**, in contrast to the threefold rotational symmetry of chiral phonons, which reverses CD between the two momentum points (see detailed discussion in Supplementary Information). Therefore, one expects a complex tangential momentum dependence of the RIXS CD at $E_3$. In fact, the RIXS CD at $E_3$ is well described by comparable amplitudes of two sinusoidal functions following either threefold (chiral phonons) or sixfold (birefringence) rotational symmetry (see Fig. 3i and Table S1). Our main discussion hereafter focuses on the modes at $E_1$ and $E_2$ unless otherwise stated.

Since CD on a phonon peak originates from angular momentum transfer between a circularly polarized photon and a chiral phonon, it could also appear for cycloidal (i.e., two-dimensional chiral) phonons[8]. Due to the symmetry requirements, only **J** perpendicular to the $c$ glide plane, i.e., cycloidal phonons, can be finite at **$q_2$**, as found in the plots of phonon chirality, defined as **J·q** (see along **$q_2$** in Fig. 3d–f), and three-dimensional arrow plots of **J** (see Fig. S2a–c in Supplementary Information) in reciprocal space obtained from DFT calculations. The absence of CD signals at **$q_2$** indicates that the CD in our RIXS measurements is not sensitive to cycloidal phonons. This is likely because **J** given by a cycloidal phonon is perpendicular to the angular momentum of an incident circularly polarized X-ray photon in our experimental geometry, inhibiting an angular momentum transfer between the photon and cycloidal phonon (see Fig. S8 and Supplementary Information for details). This argument about the geometry being insensitive to cycloidal phonons also applies to the other momentum points where we have collected RIXS, making RIXS CD sensitive solely to chiral phonons.

The RIXS CD reversal between **$q_1$** and **$q_3$** at $E_1$ and $E_2$ (see Fig. 2b, d) is consistent with the plots of **J·q** (see Fig. 3d, e) and three-dimensional arrow plots of **J** (see Fig. S2a, S2b in Supplementary Information). However, the polar plots of RIXS CD displayed in Fig. 3g, h reveal a small but finite sixfold sinusoidal contribution (birefringence) at these energy loss points. The amplitudes of the threefold sinusoidal contribution (chiral phonons) are significantly larger by approximately a factor of three than those of the sixfold sinusoidal contribution, unlike at $E_3$ (see Table S1 in Supplementary Information). This indicates that,

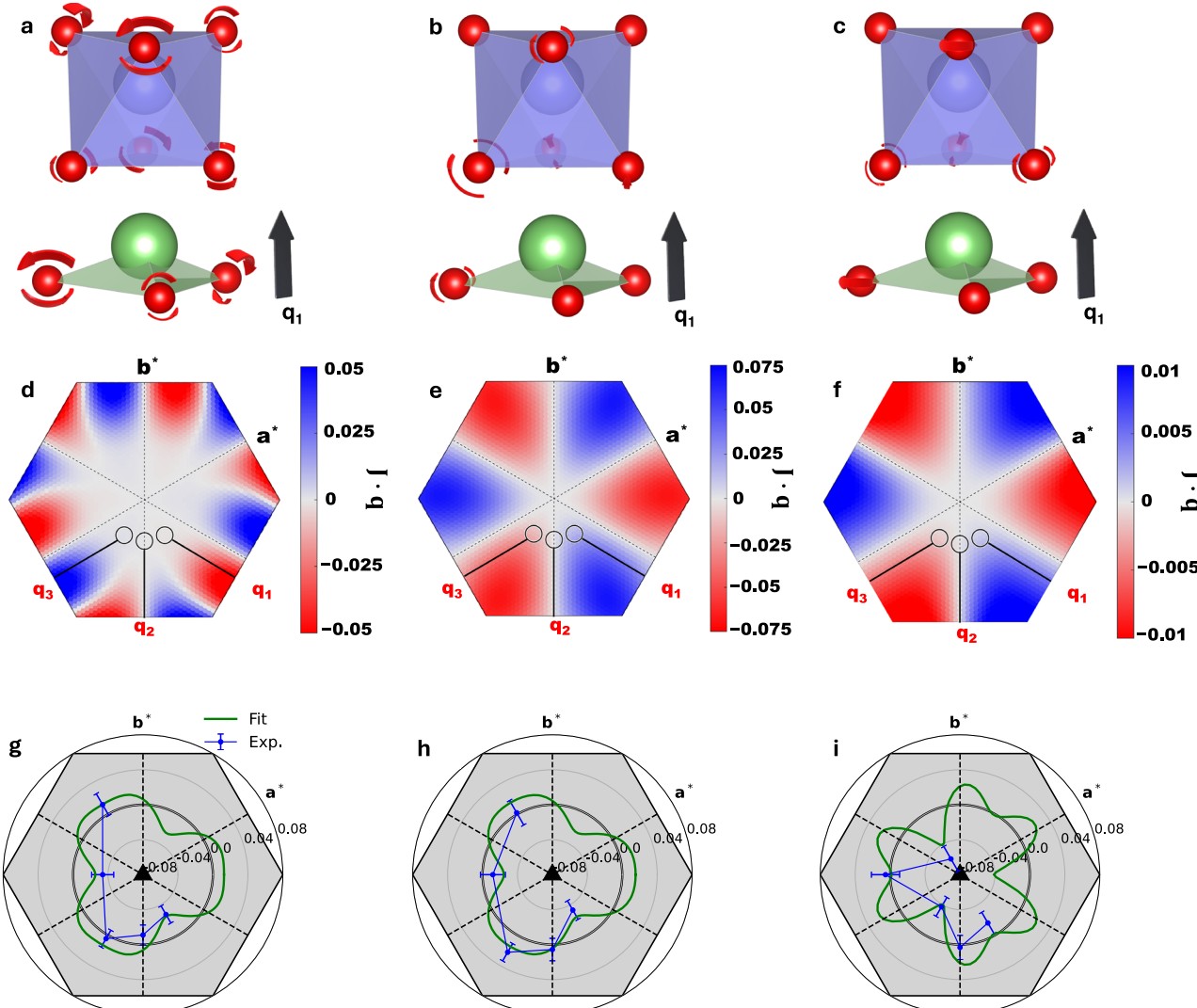

**Fig. 3 | Plots of phonon chirality and polar plots of circular dichroism in RIXS.** Chiral phonon modes at $\mathbf{q}_1$ = (0.1, 0.2, 1) at **a** -28 meV, **b** -76 meV, and **c** -105 meV, showing the main chiral revolutions of the oxygen atoms (red). We extracted only a unit composed of a $LiO_3$ pyramid and a $NbO_6$ octahedron from the unit cell for better visualization. Surface plots of phonon chirality ($\mathbf{J} \cdot \mathbf{q}$) centered at **d** $E_1$, **e** $E_2$, and **f** $E_3$, shown for the plane corresponding to ($h\,k\,1$) in hexagonal coordinates in reciprocal space. The plots consider contributions from each phonon weighted by a Gaussian centered at $E_1$, $E_2$, or $E_3$, with full width at half-maximum of 23 meV to account for the instrumental resolution. Polar plots of the RIXS CD for phonon modes at **g** $E_1$, **h** $E_2$, and **i** $E_3$. The green curves are fits with two sinusoidal functions that follows either the threefold rotational symmetry (chiral phonons) or the six-fold rotational symmetry (birefringence). Fit parameters are summarized in Table S1. The RIXS CD data points are obtained by integrating the RIXS intensity over the highlighted energy range shown in Fig. 2. The error bars are propagated from RIXS spectra with opposite circular X-ray polarization with the error bars being the standard deviation of the individual scans.

while birefringence may contribute at $E_1$ and $E_2$, it is insignificant for these phonon modes compared to $E_3$. Note that despite the considerable birefringence contribution at $E_3$, the finite contribution from the sinusoidal term with the threefold rotational symmetry, as found in Fig. 3i and Table S1, indicates that the mode at $E_3$ is also chiral. In fact, small but finite phonon chirality is predicted at $E_3$ by the DFT calculations, as shown in Figs. 3f and S2c.

Figure 3a–c visualizes the revolution components of oxygen atoms in the individual phonon modes at $\mathbf{q}_1$ (see Supplementary Videos 1–3 for the full eigenmodes at the respective energies at $\mathbf{q}_1$ viewed along $c^*$). All of them are circularly polarized, and their propagation involves the normal direction of the rotation plane, as also evident from the plots of $\mathbf{J} \cdot \mathbf{q}$ (see Fig. 3d–f). Therefore, these phonons are clearly chiral. Note that a smaller projected $\mathbf{J} \cdot \mathbf{q}$ amplitude at $E_1$ than $E_2$ (compare Fig. 3d, e) despite clearer circularly polarized eigendisplacements for the mode at $E_1$ than $E_2$ (compare Supplementary Videos 1 and 2) is due to the convolution of a large number of bands

with opposite chirality close in energy, as found in Fig. S3 in Supplementary Information. While Raman scattering with circular polarization[11,30] cannot detect g-wave chiral phonons due to the node along [001], RIXS with circular polarization provides access to chiral phonons even in such nodal systems. Note that due to the low symmetry along the employed momentum directions, neither angular momentum nor pseudo-angular momentum is a good quantum number. This accounts for the opposite relative signs between $\mathbf{J} \cdot \mathbf{q}$ and RIXS-CD signals observed between $E_1$ and $E_2$. Since circularly polarized photons can transfer angular momentum to the phononic system up to $2\hbar$ while the maximum real angular momentum of a phonon is $\hbar$, some Umklapp process can occur, as discussed in the Raman scattering process[11].

As in the case of α-quartz[12], the revolution of atoms distorts the O $2p$ electric quadrupoles by changing the Li-O-Nb bond angle, which gives rise to the sensitivity of RIXS to these phonon modes and its CD signals (see Fig. S6 showing mode effective charge of individual

phonon bands). The induced effective magnetic moment from dynamical multiferroicity, considering only the circular motions of the Born effective charges, is in the order of sub-nuclear magneton, as shown in Fig. S4 in Supplementary Information, which is similar to that found in α-quartz[12] and previous DFT calculations[15]. Recently, a large effective phononic magnetic field (~5 T) coexisting with an electronic effect (~8 T) was reported when driving the lowest-energy phonon-polaritons in $LiNbO_3$ with a circularly polarized THz pulse[41]. Assuming a similar mass magnetic susceptibility as $SrTiO_3$, ~$10^{-7}$ $cm^3/g$[42], the effective phononic magnetic field can be explained when the phononic magnetization is ~$1\mu_n$ (nuclear magneton) per unit cell, as consistent with the previous calculations[15] and ours.

In conclusion, we have demonstrated the direct observation of chiral phonons in a polar crystal. Chiral phonons are a key ingredient of recently discovered exotic phenomena due to their phonon angular momentum[24–26], magnetism mediating the spin-lattice coupling[23,28,43], and non-trivial coupling with electrons[27]. Although $LiNbO_3$ itself does not exhibit significant spin-lattice coupling, phonon angular momentum has been discussed in α-quartz[25], where such coupling is also negligible. More importantly, our demonstration of chiral phonons in a polar crystal establishes their presence across a wide range of polar materials, including those where spin-lattice coupling is essential in emergent physics. Switching ferroelectric domains is straightforward in contrast to switching chiral domains. Such domain inversion allows us to switch angular momentum and therefore the spin-lattice coupling in-situ at general momentum points. This enables unique opportunities to explore physical properties based on controllable phonon chirality or its associated phonon angular momentum, and hence, opens the door for chiral phononics. Based on the bistability of ferroelectric domains and the developed technology for local switching thereof[31], patterning chiral phononic devices in nanoscales will be possible. Another perspective is the possible phonon angular momentum switching at ultrafast timescales due to the ultrafast reversal of ferroelectric polarization[44]. The electrically controllable bistable states of phonon chirality or phonon angular momentum have significant potential to tailor emergent phenomena associated with chiral phonons.

## Methods

### RIXS

RIXS measurements were performed at Beamline I21 at the Diamond Light Source in the UK[45]. We tuned the photon energy to around the O K edge. The energy resolution was estimated as ~23 meV full width at half-maximum from the elastic peak of a carbon tape. A $LiNbO_3$ crystal with a single ferroelectric domain state and the largest face perpendicular to c in the hexagonal setting was commercially purchased. The manipulator at the beamline allows us to access different momentum points during the experiment. The error bars in an RIXS spectrum are the standard deviation of individual scans. The X-ray absorption spectrum was obtained by the partial fluorescence yield before the RIXS measurements.

### DFT

Density functional perturbation theory calculations of the phonon frequencies and eigenvectors of $LiNbO_3$ were performed using the Abinit software package (v. 10)[46]. The calculations used the PBE GGA exchange–correlation functional[47] with the vdw-DFT-D3(BJ) dispersion correction[48]. The PAW method was used with a plane-wave basis set cutoff energy of 150 Ha within the PAW spheres and 30 Ha without. PAW basis sets were used as received from the Abinit library. A $5 \times 5 \times 5$ Monkhorst–Pack grid[49] was used to sample both **k**-points and **q**-points. The **k**-point grid spacing and plane-wave basis set cutoff energy were chosen following convergence studies, with the convergence criterion being 1% in pressure. Prior to the phonon calculation, the structure was relaxed to an internal pressure of −9 MPa, resulting in hexagonal lattice

constants of $a = 5.13$ Å and $c = 13.9$ Å, in good agreement with experimental values ($a = 5.14$ Å and $c = 13.8$ Å)[50]. Calculations of the phonon circular polarization vectors and phonon magnetic moments were performed using a MATLAB script that is available as supplemental data.

## Data availability

Experimental and model data are accessible from the PSI Public Data Repository[51].

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

## Acknowledgements

The resonant inelastic X-ray scattering experiment was performed at beamline I21 at the Diamond Light Source (proposal MM36210). A.N. acknowledges funding from the Swiss National Science Foundation through Project No. 20021-196964. C.P.R. acknowledges support from the project FerrMion of the Ministry of Education, Youth and Sports, Czech Republic, co-funded by the European Union (CZ.02.01.01/00/22_008/0004591), the European Union and Horizon 2020 through grant no. 810451, and ETH Zurich. Computational resources were provided by the Swiss National Supercomputing Center (CSCS) under project ID s1128.

## Author contributions

H.U., A.N., and U.S. conceived and designed the project. H.U., A.N., M.G.F., K.-J.Z., and U.S. performed resonant inelastic X-ray scattering experiments. A.N. analysed the experimental data. C.P.R. performed density-functional theory calculations. H.U., A.N., and U.S. interpreted the experimental data. H.U., A.N., C.P.R., and U.S. wrote the manuscript with contributions from all authors.

## Competing interests

The authors declare no competing interests.
