## [Transparent Peer Review file · Nature Communications]

Chiral phonons in polar LiNbO₃

Corresponding Author: Dr Hiroki Ueda

Version 0:

Reviewer comments:

Reviewer #1

(Remarks to the Author)

The authors of NCOMMS-25-67448-T report an oxygen K-edge resonant inelastic x-ray scattering (RIXS) study on a prototypical polar LiNbO₃ crystal. They observed circular dichroism (CD) in phonon excitations and ascribe it to phonon chirality.

Chiral phonons have attracted considerable interest in recent years because their angular momentum can couple to magnetism, leading to intriguing physical phenomena described in detail in the introduction of the manuscript. Some of the authors of this manuscript previously reported the observation of chiral phonons in alpha-quartz (Ref. 12). Nevertheless, observations of chiral phonons remain rare and experimentally challenging. Moreover, observations of chiral phonons that fulfill the symmetry requirement of being dynamic chiral objects have so far been limited to chiral crystals.

The existence of chiral phonons in non-centrosymmetric crystals has been predicted theoretically (Ref. 32), implying that they need not exist only in chiral crystals. The argument at the bottom of page 3 of the manuscript builds on this theory. An experimental attempt to detect chiral phonons in an achiral crystal is therefore noteworthy. Furthermore, the authors claim that symmetry considerations lead to nodes in the CD in momentum space, in stark contrast to the case of the chiral alpha-quartz crystal. [I note that symmetry of the nodes of chiral phonons was systematically studied in a recent preprint (Y. Yang et al., arXiv:2506.13721v1).] If successfully demonstrated, it will have a significant impact on the field of condensed matter physics.

However, experimental evidence presented in the current manuscript is unfortunately weak and inconclusive. At first glance at the difference spectra in Fig. 2b-d, distinguishing CD signals from noises is not straightforward. In Fig. 3g, the CD magnitude is nearly zero at q₃, while at q₄ it is about twice as large as the amplitude of the fits. This means that the discrepancy is comparable to the oscillation amplitude in the a*b*-plane (and the experimental error bars). A similar issue arises in Fig. 3h, where the CD magnitude is nearly zero at q₅ and twice the amplitude of the fits at q₁. At the very least, the authors should provide the spectra at q₄ and q₅ for a fair assessment of the data. A semi-quantitative evaluation of birefringence might help to mitigate this discrepancy. Alternatively, if a microscopic argument is presented regarding the relationship between the sign of J_{dot}q in Figs. 3d-f and the sign of the CD, then qualitative agreement between J_{dot}q and CD could be acceptable. Comparing at q₁, J_{dot}q is negative at E1 and positive at E2, whereas the CD is negative at both E1 and E2.

In addition, I suggest the authors address the following minor points:

- a) What energy range was integrated for the points shown in Fig. 3g-i?
- b) If the reason for choosing LiNbO₃ is that a single-domain crystal can be obtained, please state this explicitly for the reader.

In conclusion, this manuscript tackles an important and timely topic with the potential for strong impact, but experimental evidence is not convincing enough to meet high standards of Nature Communications. I consider that a more careful analysis of the data or a detailed microscopic argument is necessary. Therefore, I do not recommend the manuscript for the publication in its present form.

Reviewer #2

(Remarks to the Author)
Dear Editor

This work reports a RIXS study of phonon modes in LiNbO₃. The main claim is that some of those modes are chiral, as inferred from RIXS spectra with circular X-ray polarisation. This is a nice work, which should be published in some journal but I'm not sure NatureComm is best suited.

Although very well documented, the introduction is difficult to read, and positioning of the paper is not that obvious. What is the point? For example, the authors write "... chiral phonons have only been experimentally reported in chiral crystals, limiting the exploration of chiral phononics. Here, we demonstrate the presence of chiral phonons in the prototypical polar Crystal LiNbO₃ within a single ferroelectric domain by using resonant inelastic X-ray scattering (RIXS) with circular polarization". What is the consequence of this result? This is hardly explained in the paper. Why is it important to a non specialized audience?

If DFT predicts the existence of such chiral modes in LiNbO₃, then, again, what is the point? It is also very likely that DFT is also able to calculate the lattice dynamics in a bunch of materials, hence to identify chiral phonons in many other systems?

Furthermore, the authors write "Three different types of phonons have been referred to as chiral phonons in the Community ... a rotational mode that propagates perpendicular to the rotation plane, such as the one observed in chiral crystals, e.g., α -quartz [12], α -HgS [11], and tellurium [29]. All of them possess phonon angular momentum, but only the last type fulfills the symmetry requirements [30] of being a dynamical chiral object. Here again, what is the point? To explain that among these different types, only the third one is correct? In this case, much more explanations are needed, referring to [30] being too short

In the conclusion, the authors tend to clarify the motivation of the paper: "As chiral phonons are a key ingredient of recently discovered exotic phenomena due to their phonon angular momentum [24-26], magnetism mediating the spin-lattice coupling [23,28,40], and non-trivial coupling with electrons [27], the demonstration of chiral phonons in a polar crystal is of significant importance as it opens the door for chiral phononics. I understand that opening the door might be important, but the physics of spin-lattice coupling is so far from LiNbO₃ that, to me, the present work appears more as a beautiful experiment rather than an important step in this particular field. Therefore, I'm not inclined to support publication unless the authors make an effort to explain the physical condition driving the rise of chiral phonons, how DFT helps, why LiNbO₃ is relevant, and clearly explain their point.

Version 1:

Reviewer comments:

Reviewer #1

(Remarks to the Author)

The authors have carefully considered my previous comments, and the revision of the manuscript is satisfactory. Although the signal-to-noise ratio is close to the detection limit, I accept the authors' claim. Therefore, I recommend the revised manuscript for publication in Nature Communications.

As an additional suggestion, the authors may consider citing a recent review paper on chiral phonons [D. M. Juraschek et al., Nat. Phys. 21, 1532 (2025)], which provides a useful discussion distinguishing three different types of phonons that have been referred to as chiral phonons.

Reviewer #2

(Remarks to the Author)

Dear Editor

Thank you for this new version. Upon rereading it, I think I misjudged this manuscript. The introduction actually provides a detailed explanation of many phenomena related to these chiral phonons, and I regret not having better understood the scope of this paper. It is now much clearer to me that observing a chiral phonon in a material with a polar space group is important, firstly because it is much less restrictive than the same observation reduced to chiral materials, and secondly because this class of compounds includes some examples of multiferroic compounds, for example. Similarly, the combination of DFT calculations and experiments is important, and I can only regret that the (highly educational) videos are only available in Supmat. I am therefore now convinced and recommend the publication of this paper.

In the point-by-point response below, Referee comments are presented in blue, our replies are in black, and changes made to the manuscript are indicated in red.

Referee 1:

We greatly appreciate the Referee's positive assessment of our work as an important contribution to this timely topic in condensed matter physics, as well as his/her critical evaluation of the data quality. The Referee requested more rigorous data analysis and/or a detailed microscopic argument to substantiate our findings and strengthen the manuscript to meet the high standards of *Nature Communications*. We found these comments to be insightful, constructive, and highly valuable in significantly improving the quality of the revised manuscript.

1. At first glance at the difference spectra of Fig. 2b-d, distinguishing CD signals from noises is not straightforward.

As mentioned in the main text, chiral phonon splitting in LiNbO_3 is smaller than in α -quartz due to the g -wave nature of the chiral phonons. As a result, the signal-to-noise ratio (SNR) of the RIXS-CD signals approaches the detection limit. Nevertheless, we remain confident that the observed RIXS-CD signals represent intrinsic chiral phonon signatures. Figure R1 (see below) shows the RIXS spectra with circular polarization (top), normalized RIXS circular contrast $[(I_{C-} - I_{C+})/(I_{C-} + I_{C+})]$ (middle), and the normalized RIXS circular contrast divided by standard errors (bottom), which represents SNR, at all the momentum points. While the majority of data points in regions where no circular dichroism is expected statistically lie within $|\text{SNR}| < 1$, the data points around E_1 , E_2 , and E_3 often exceed $|\text{SNR}| > 2$, demonstrating statistically significant signals.

Fig. R1 | RIXS spectra with circular polarization (top), normalized RIXS circular contrast $[(I_{C-} - I_{C+})/(I_{C-} + I_{C+})]$ (middle), and the normalized RIXS circular contrast divided by standard errors representing SNR (bottom) at all the momentum points (**a**: q_1 , **b**: q_2 , **c**: q_3 , **d**: q_4 , and **e**: q_5). Highlighted regions are $|\text{SNR}| < 1$.

2. In Fig. 3g, the CD magnitude is nearly zero at q_3 , while at q_4 it is about twice as large as the amplitude of the fits. This means that the discrepancy is comparable to the oscillation amplitude in the a^*b^* -plane (and the experimental error bars). A similar issue arises in Fig. 3h, where the CD magnitude is nearly zero at q_5 and twice the amplitude of the fits at q_1 .
3. A semi-quantitative evaluation of birefringence might help to mitigate this discrepancy. Alternatively, if a microscopic argument is presented regarding the relationship between the sign of $J_{\dot{q}}$ in Figs. 3d-f and the sign of the CD, then qualitative agreement between $J_{\dot{q}}$ and CD could be acceptable. Comparing at q_1 , $J_{\dot{q}}$ is negative at E_1 and positive at E_2 , whereas the CD is negative at both E_1 and E_2 .

These questions are closely related. The Referee correctly identifies a relatively large discrepancy between the RIXS-CD data and the fit using only the threefold term in Figs. 3g and 3h. This discrepancy is likely attributable to birefringence, as previously discussed for the data measured at E_3 (Fig. 3i). Including a sixfold symmetric term to account for the birefringence effect indeed yields improved fits for these data. Notably, while the data at E_3 show comparable amplitudes for the threefold and sixfold terms, the data at E_1 and E_2 exhibit a threefold term amplitude approximately three times larger than the sixfold term amplitude. In the revised manuscript, we have incorporated the sixfold term into the fits of RIXS-CD data at E_1 and E_2 as well, and have replaced Figs. 3g and 3h accordingly. We have also modified the descriptions of the fitting procedure on p. 7 and added the following sentences to emphasize the dominant contribution from chiral phonons with only a minor birefringence contribution at these energy losses. The newly added Table S1 summarizes all fit parameters.

“In fact, the RIXS CD at E_3 is well described by comparable amplitudes of two sinusoidal functions following either threefold (chiral phonons) or sixfold (birefringence) rotational symmetry (see Fig. 3i and Table S1).” on p. 6.

“However, the polar plots of RIXS CD displayed in Figs. 3g and 3h reveal a small but finite sixfold sinusoidal contribution (birefringence) at these energy loss points. The amplitudes of the threefold sinusoidal contribution (chiral phonons) are significantly larger by approximately a factor of three than those of the sixfold sinusoidal contribution, unlike at E_3 (see Table S1 in Supplementary Information). This indicates that, while birefringence may contribute at E_1 and E_2 , it is insignificant for these phonon modes compared to E_3 . Note that despite the considerable birefringence contribution at E_3 , the finite contribution from the sinusoidal term with the threefold rotational symmetry, as found in Fig. 3i and Table S1, indicates that the mode at E_3 is also chiral.” on p. 7.

Fig. 3 | **a-c**, Chiral phonon modes at $\mathbf{q}_1 = (0.1, -0.2, 1)$ at **a**, ~ 28 meV, **b**, ~ 76 meV, and **c**, ~ 105 meV, showing the main chiral revolutions of the oxygen atoms (red). We extracted only a unit composed of a LiO_3 pyramid and a NbO_6 octahedron from the unit cell for better visualization. **d-f**, Surface plots of phonon chirality ($\mathbf{J} \cdot \mathbf{q}$) centered at **d**, E_1 , **e**, E_2 , and **f**, E_3 , shown for the plane corresponding to $(hk1)$ in hexagonal coordinates in reciprocal space. The plots consider contributions from each phonon weighted by a Gaussian centered

at E_1 , E_2 , or E_3 , with full width at half-maximum of 23 meV to account for the instrumental resolution. **g-i**, Polar plots of the RIXS CD for phonon modes at **g**, E_1 , **h**, E_2 , and **i**, E_3 . The green curves are fits with two sinusoidal functions that follows either the threefold rotational symmetry (chiral phonons) or the sixfold rotational symmetry (birefringence). **Fit parameters are summarized in Table S1. The RIXS CD data points are obtained by integrating the RIXS intensity over the highlighted energy range shown in Fig. 2.** The error bars are propagated from RIXS spectra with opposite circular X-ray polarization with the error bars being the standard deviation of the individual scans.

“Fitting polar plots

We fit the polar plots shown in Figs. 3**g-3i** using two sinusoidal functions: one with threefold symmetry (amplitude: a_3) representing the chiral phonon contribution, and one with sixfold symmetry (amplitude: a_6) representing the birefringence effect. Table S1 summarizes the fit parameters.” in Supplementary Information.

Table S1 | Fit amplitudes for the polar plots of RIXS-CD signals shown in Figs. 3**g-3i** ($E_1 - E_3$).

	E_1	E_2	E_3
a_3	$(1.9 \pm 0.6) \times 10^{-2}$	$(2.0 \pm 0.7) \times 10^{-2}$	$(-2.0 \pm 0.7) \times 10^{-2}$
a_6	$(-6 \pm 5) \times 10^{-3}$	$(-5 \pm 6) \times 10^{-3}$	$(-2 \pm 0.5) \times 10^{-2}$

The Referee raised an important question regarding the relation between the signs of $\mathbf{J} \cdot \mathbf{q}$ (phonon chirality) and the RIXS-CD signals. In crystals with discrete rotation symmetry C_n , continuous angular momentum does not commute with the Hamiltonian and is therefore not a conserved quantity. Instead, pseudo-angular momentum with modulus equal to the rotation symmetry order n serves as a good quantum number. Consequently, phonon angular momentum ($\mathbf{J}_{\text{phonon}}$) is often represented using pseudo-angular momentum. During photon-phonon scattering processes along high-symmetry directions with rotation symmetry, the sum of photon angular momentum and the pseudo-angular momentum of phonons is conserved. Therefore, RIXS-CD signals, which indicate more efficient photon angular momentum transfer for one helicity over the other, should exhibit a one-to-one correspondence with the pseudo-angular momentum remaining in the phonons.

However, along directions without rotation symmetry, pseudo-angular momentum is not a good quantum number. Therefore, we use angular momentum to represent $\mathbf{J}_{\text{phonon}}$ for the surface plots shown in Figs. 3**d-3f**. Although angular momentum transfer can still occur

during photon-phonon scattering processes, as evidenced by the observed finite RIXS-CD signals, it is not physically meaningful to seek a direct and general relation between the signs of $\mathbf{J}_{\text{phonon}}$ and RIXS-CD signals in these cases. Although we cannot predict the sign of the RIXS-CD signals at a given momentum point, the symmetry-related points have a clear relation in sign and amplitude, as observed (see polar plots in Figs. 3g-3i). Note that while circularly polarized photons can transfer angular momentum of $2\hbar$ to the phononic system, the maximum real angular momentum of a phonon is \hbar , indicating the occurrence of the Umklapp process (as discussed in Ref. 11). In the revised manuscript, we have added the following sentence on p. 9 to address this point.

“Note that due to the low symmetry along the employed momentum directions, neither angular momentum nor pseudo-angular momentum is a good quantum number, even though angular momentum transfer can still occur. This accounts for the opposite relative signs between $\mathbf{J}\cdot\mathbf{q}$ and RIXS-CD signals observed between E_1 and E_2 . Since circularly polarized photons can transfer angular momentum to the phononic system up to $2\hbar$ while the maximum real angular momentum of a phonon is \hbar , some Umklapp process can occur, as discussed in the Raman scattering process [11].”

4. At the very least, the authors should provide the spectra at \mathbf{q}_4 and \mathbf{q}_5 for a fair assessment of the data.

Following the Referee’s suggestion, we have included the RIXS spectra at \mathbf{q}_4 and \mathbf{q}_5 as Fig. S9 in the Supplementary Information. Equivalent spectra are shown in Figs. R1d and R1e.

“RIXS spectra at \mathbf{q}_4 and \mathbf{q}_5

RIXS spectra collected at \mathbf{q}_4 and \mathbf{q}_5 , which were used to generate the data shown in Fig. 3, are shown in Fig. S9.”

Fig. S9 | RIXS spectra taken with circular polarizations at **a**, **q₄** and **b**, **q₅**.

5. What energy range was integrated for the points shown in Fig. 3g-i?

We integrated the intensities over the energy range indicated by the green highlighted bars in Figs. 2b-2d. To clarify this point, we have added the following sentence to the caption of Fig. 3.

“The RIXS CD data points are obtained by integrating the RIXS intensity over the highlighted energy range shown in Fig. 2.”

6. If the reason for choosing LiNbO₃ is that a single-domain crystal can be obtained, please state this explicitly for the reader.

We selected LiNbO₃ for several reasons. To clarify the rationale for this choice, we have added the following sentence to the Introduction section:

“LiNbO₃ was selected for this study for several reasons: (1) it is a well-established prototypical polar crystal, (2) high-quality single-domain crystals are commercially available, (3) ferroelectric domains can be controlled at the nanometer scale [31], and (4) its phonon properties are directly relevant to technological applications, such as substrates for surface acoustic wave devices, thin-film growth, and piezoelectric devices.” on p. 3

7. Furthermore, the authors claim that symmetry considerations lead to nodes in the CD in momentum space, in stark contrast to the case of the chiral alpha-quartz crystal. [I note that symmetry of the nodes of chiral phonons was systematically studied in a recent preprint (Y. Yang et al., arXiv:2506.13721v1).] If successfully demonstrated, it will have a significant impact on the field of condensed matter physics.

We thank the Referee for bringing this valuable recent preprint to our attention, which provides a systematic overview of nodal chiral phonons. We have added this reference to the revised manuscript and modified the following sentence to classify the observed phonons according to the symmetry categories proposed in the preprint.

37. Yang, Y., Xiao, Z., Mao, Y., Li, Z., Wang, Z., Deng, T., Tang, Y., Song, Z.-D., Li, Y., Yuan, H., Shi, Ming, and Xu, Y., Catalogue of chiral phonon materials. *arXiv*: 2506.13721. DOI: <https://doi.org/10.48550/arXiv.2506.13721>

“Recent symmetry-based theoretical work categorized chiral phonons in crystals with C_{3v} symmetry as g-wave because of the nodal structure [37].” on p. 6.

8. In conclusion, this manuscript tackles an important and timely topic with the potential for strong impact, but experimental evidence is not convincing enough to meet high standards of *Nature Communications*. I consider that a more careful analysis of the data or a detailed microscopic argument is necessary.

As summarized in our responses to the Referee’s comments above, we have carefully re-evaluated the signal-to-noise ratio of our RIXS-CD signals and incorporated a quantitative analysis of the birefringence contribution in the revised manuscript. Furthermore, we have emphasized the significance of our work as the first experimental detection of g-wave chiral phonons, as the Referee noted (see also our reply to the 1st to 3rd points by the Second Referee), and added a detailed discussion on pseudo-angular momentum and angular momentum transfer. We believe these substantial revisions have made the manuscript sufficiently robust and convincing to meet the high standards of *Nature Communications*, and we respectfully hope that the Referee will now support its publication.

Referee 2:

We sincerely thank the Referee for recognizing the quality of our experimental results. The Referee requested clarifications regarding the positioning of our work within the broader community context, as well as improvements to enhance the clarity of several key points.

1. Although very well documented, the introduction is difficult to read, and positioning of the paper is not that obvious. What is the point ? For example, the authors write "... chiral phonons have only been experimentally reported in chiral crystals, limiting the exploration of chiral phononics. Here, we demonstrate the presence of chiral phonons in the prototypical polar Crystal LiNbO₃ within a single ferroelectric domain by using resonant inelastic X-ray scattering (RIXS) with circular polarization". What is the consequence of this result ? This is hardly explained in the paper. Why is it important to a non specialized audience ?

2. If DFT predicts the existence of such chiral modes in LiNbO₃, then, again, what is the point? It is also very likely that DFT is also able to calculate the lattice dynamics in a bunch of materials, hence to identify chiral phonons in many other systems?
3. In the conclusion, the authors tend to clarify the motivation of the paper: "As chiral phonons are a key ingredient of recently discovered exotic phenomena due to their phonon angular momentum [24-26], magnetism mediating the spin-lattice coupling [23,28,40], and non-trivial coupling with electrons [27], the demonstration of chiral phonons in a polar crystal is of significant importance as it opens the door for chiral phononics. I understand that opening the door might be important, but the physics of spin-lattice coupling is so far from LiNbO₃ that, to me, the present work appears more as a beautiful experiment rather than an important step in this particular field.

These points are interconnected and concern the motivation and positioning of our manuscript within the broader research community and non-specialist readers. First and foremost, the central message of our study is the experimental demonstration of chiral phonons in a polar crystal. This expands the scope of chiral phononics from chiral crystals to the much broader class of non-centrosymmetric materials and significantly enhances its potential for *in-situ* control of phonon chirality and associated phonon angular momentum, a capability unavailable in chiral crystals (as mentioned in p. 10). While DFT calculations can predict chiral phonons and other interesting properties in many materials, experimental verification is essential to establish their existence unambiguously and to confirm the underlying symmetry principles predicted by theory. The direct detection beyond a single example (α -quartz) to a new class of materials (polar crystals or even more general non-centrosymmetric materials) will further stimulate the study of their role in emergent phenomena.

As explained in our response to the 6th point raised by the First Referee, we selected LiNbO₃ for this study because: (1) it is a well-established polar crystal, (2) high-quality single-domain crystals are readily available, (3) ferroelectric domains can be controlled at the nanometer scale, and (4) its phonon properties have direct technological relevance. Although LiNbO₃ itself does not exhibit strong spin-lattice coupling, such coupling plays an essential role in the emergent physics of various ferroelectric systems, such as multiferroics and ferroelectrics/heavy-element interfaces. Our experimental demonstration, supported by DFT calculations, establishes the relevance of chiral phonons in polar crystals more broadly, including those with spin-lattice coupling. It is worth noting that phonon angular momentum has, for instance, been discussed in α -quartz (Ref. 25), where spin-lattice coupling is also

negligible. Therefore, chiral phonons can serve as a key emergent ingredient in materials even without strong spin-lattice coupling, like LiNbO₃. While we sincerely appreciate the Referee's acknowledgment of our work as a beautiful experiment, we believe our experimental demonstration of chiral phonons in a polar crystal has implications extending well beyond LiNbO₃. It represents an essential step forward in condensed matter physics, as explicitly mentioned by the first Referee.

Moreover, our experimental approach overcomes technical limitations that prevent other techniques from accessing these chiral phonon modes. It is particularly noteworthy that optical techniques previously employed to measure "chiral" phonons, such as Raman scattering with circular polarization, cannot be applied to LiNbO₃ due to strong birefringence in the optical range, which limits the accessible momentum direction along the principal axis where a node exists for g-type chiral phonons (as categorized in Ref. 37). This further underscores the value and necessity of our experimental approach.

To clarify the positioning and significance of our work, we have added the following sentences to the manuscript:

"Density functional theory (DFT) calculations provide reliable phonon properties for LiNbO₃ because it contains only a moderate number of non-magnetic light elements in the unit cell without significant electronic correlations. Therefore, combining RIXS experiments and DFT calculations yields an unambiguous and robust demonstration of chiral phonons in a polar (non-chiral) crystal and establishes a large and important class of materials for chiral phononics." on p. 3.

"Chiral phonons are a key ingredient of recently discovered exotic phenomena due to their phonon angular momentum [24-26], magnetism mediating the spin-lattice coupling [23,28,41], and non-trivial coupling with electrons [27]. Although LiNbO₃ itself does not exhibit significant spin-lattice coupling, phonon angular momentum has been discussed in α -quartz [25], where such coupling is also negligible. More importantly, our demonstration of chiral phonons in a polar crystal establishes their presence across a wide range of polar materials, including those where spin-lattice coupling is essential in emergent physics. Switching ferroelectric domains is straightforward in contrast to chiral domains. Such domain inversion allows us to switch angular momentum and therefore the spin-lattice coupling *in-situ* at general momentum points. This enables unique opportunities to explore physical

properties based on controllable phonon chirality or its associated phonon angular momentum, **thereby opening the door for chiral phonics.**” on p. 10.

“Raman spectroscopy with circular polarization probed chiral phonons propagating only along the principal axis, which is the c axis in trigonal crystals, because of the strong birefringence effect [11,29]. However, it does not measure chiral phonons in LiNbO_3 because the c axis is in glide planes. **RIXS-CD measurements can overcome this limitation by their large momentum accessibility to extend a range of materials relevant for chiral phonics.**” on p. 5.

“**While Raman scattering with circular polarization [11,29] cannot detect g-wave chiral phonons due to the node along [001], RIXS with circular polarization provides access to chiral phonons even in such nodal systems.**” on p. 9.

4. Furthermore, the authors write "Three different types of phonons have been referred to as chiral phonons in the Community ... a rotational mode that propagates perpendicular to the rotation plane, such as the one observed in chiral crystals, e.g., α -quartz [12], α -HgS [11], and tellurium [29]. All of them possess phonon angular momentum, but only the last type fulfills the symmetry requirements [30] of being a dynamical chiral object. Here again, what is the point ? To explain that among these different types, only the third one is correct ? In this case, much more explanations are needed, referring to [30] being too short.

The Referee noted that our manuscript distinguishes three different types of rotational phonons that have been referred to as “chiral phonons” in the literature: (1) those limited to the Γ point, (2) those propagating within the rotation plane, and (3) those propagating perpendicular to (or with a component perpendicular to) the rotation plane. The Referee requested clarification regarding the purpose of this classification.

Symmetry classification is fundamental to understanding material properties. As stated, all these types of phonons carry angular momentum and therefore break time-reversal symmetry for a given momentum, which can be relevant with respect to magnetic properties. However, when considering the reciprocal lattice volume, (3) truly chiral phonons, that is, rotational phonon modes that have propagation including the perpendicular component to the rotational plane, are the most relevant to macroscopic properties, such as transport, among the three types of phonons that have been referred to as “chiral” phonons in the literature. This is

because phonons confined to the two-dimensional plane (1) at the Γ point or (2) within the rotational plane occupy a negligible fraction of momentum space compared to phonons that extend throughout three-dimensional reciprocal space.

In response to this comment, we have modified and expanded the explanations, as follows.

“All of them possess phonon angular momentum, but **the reciprocal lattice volume where these types of phonons exist is significantly larger for the last type compared to the other two because the former two types are confined to a two-dimensional plane, whereas the last type of phonons reside in three-dimensional space. This distinction can lead to different contributions to macroscopic properties. Furthermore, from a symmetry perspective [30], only the last type fulfills the symmetry requirements of being a dynamical chiral object due to these different dimensionalities.** Hence, **the most relevant and truly** chiral phonons have only been experimentally reported in chiral crystals, limiting the exploration of chiral phononics.”
on p. 3.

5. **Therefore, I'm not inclined to support publication unless the authors make an effort to explain the physical condition driving the rise of chiral phonons, how DFT helps, why LiNbO₃ is relevant, and clearly explain their point.**

Through the substantial revisions summarized above, we have addressed each of these concerns by clarifying: (1) the rationale for selecting LiNbO₃ for this study, (2) how DFT calculations support and validate our experimental findings within the broader chiral phononics communities, and (3) the critical importance of experimentally demonstrating truly chiral phonons in non-chiral crystals. We believe these revisions have significantly strengthened the manuscript, and we respectfully hope that the Referee will now support its publication.

Below, Referee comments are presented in blue, our replies are in black, and changes made to the manuscript are indicated in red.

Referee 1:

We sincerely appreciate the Referee's careful review and recommendation of our manuscript for its publication. The Referee raised a reference for consideration of citation.

1. As an additional suggestion, the authors may consider citing a recent review paper on chiral phonons [D. M. Juraschek et al., *Nat. Phys.* **21**, 1532 (2025)], which provides a useful discussion distinguishing three different types of phonons that have been referred to as chiral phonons.

Following the Referee's suggestion, we have added this reference to the revised manuscript.

29. Juraschek, D. M., Geilhufe, R. M., Zhu, H., Basini, M., Baum, P., Baydin, A., Chaudhary, S., Fechner, M., Flebus, B., Grissonnanche, G., Kirilyuk, A. I., Lemeshko, M., Maehrlein, S. F., Mignolet, M., Murakami, S., Niu, Q., Nowak, U., Romao, C. P., Rostami, H., Satoh, T., Spaldin, N. A., Ueda, H., and Zhang, L., Chiral phonons. *Nat. Phys.* **21**, 1532-1540 (2025). DOI: <https://doi.org/10.1038/s41567-025-03001-9>